# Community-Based Workshops to Involve Rural Communities in Wildlife Management Case Study: Bighorn Sheep in Baja California, Mexico

**DOI:** 10.3390/ani13203171

**Published:** 2023-10-11

**Authors:** Enrique de J. Ruiz-Mondragón, Guillermo Romero-Figueroa, Rafael Paredes-Montesinos, Luz A. Tapia-Cabazos, Luis A. Méndez-Rosas, Crystian S. Venegas-Barrera, María E. Arrellano-García, Israel Guerrero-Cárdenas, Eloy A. Lozano-Cavazos

**Affiliations:** 1Facultad de Ciencias, Universidad Autónoma de Baja California, Ensenada 22860, BC, Mexico; ruize56@uabc.edu.mx (E.d.J.R.-M.); rparedes90@uabc.edu.mx (R.P.-M.); luz.tapia@uabc.edu.mx (L.A.T.-C.); luis.mendez93@uabc.edu.mx (L.A.M.-R.); evarista.arellano@uabc.edu.mx (M.E.A.-G.); 2Tecnológico Nacional de México, Benito Juárez 03330, CDMX, Mexico; crystian.vb@cdvictoria.tecnm.mx; 3División de Estudios de Posgrado e Investigación, Instituto Tecnológico de Ciudad Victoria, Ciudad Victoria 87010, TAMPS, Mexico; 4Centro de Investigaciones Biológicas del Noroeste, La Paz 23096, BCS, Mexico; guerrero04@cibnor.mx; 5Departamento de Recursos Naturales Renovables, Universidad Autónoma Agraria Antonio Narro, Saltillo 25315, COAH, Mexico; alejandrolzn@yahoo.com

**Keywords:** community-based conservation, community-based management, community-based monitoring, ethnozoology, participatory mapping, rural capacity building

## Abstract

**Simple Summary:**

Sustainable wildlife management is achieved when planning is based on a thorough knowledge of the species to be harvested. The objective of this research was to generate such knowledge in two rural communities in Mexico through a collaborative process that integrates formal and traditional knowledge about bighorn sheep management. A program of community workshops was implemented in which the communities discussed their interest in bighorn sheep, trained villagers in their management, described their habitat, and planned their monitoring. During the workshops, the communities identified their primary interest in the species as economic, identified the main factors threatening its conservation, created a detailed map of its habitat, and designed the strategy they would use to monitor its population. The workshop program proposed in this paper aims to train rural communities, generate relevant information for the management of wild species, and lay the foundation for a long-term conservation project.

**Abstract:**

The description of natural history, and information on the factors threatening conservation, the distribution area, and the status of species population are necessary for proper wildlife management. The objective of this research was to generate such information in two rural communities and to engage residents in bighorn sheep management through a program of three workshops. The first one covered training regarding natural history and management of the species. The second one consisted in the description of the habitat of the species through a dynamic of participatory mapping. The third, include a design of a one strategy to monitor the bighorn sheep population. The workshops were attended by 37 people from the two rural communities. The results suggest the economic element was the main interest of the inhabitants regarding the bighorn sheep. Eleven risk factors were identified to the bighorn sheep in the study sites, a participatory map with relevant information for the management of the species on each community was developed, and a monitoring strategy of the bighorn sheep population was prepared. The workshop program proposed in this research is a tool that can be applied in rural communities to lay the groundwork for a long-term management project of wildlife species.

## 1. Introduction

Biodiversity conservation refers to the preservation of genes, species, and different types of ecosystems [1]; and since human activities destroy these three elements, biodiversity conservation necessarily implies the coexistence of people with nature [2] (pp. 1–14). Hence, community engagement is extremely important for biodiversity conservation, because to protect natural resources it is necessary to work with people, their lives, their aspirations, their fears, and the complexity of their society [1,3].

Community-based conservation programs emerged as a mean to engage local people in the preservation of natural resources found on lands they own [4,5,6] and are recognized as one of the major global forces for the protection and sustainable management of natural resources [7]. The goal of community-based conservation is to improve human well-being and conserve biodiversity through community development initiatives [5]. This is achieved by empowering people living in natural areas to participate in land-use policy and management decisions; by giving people ownership of wildlife resources; and by providing local people with economic benefits from biodiversity conservation [4].

The community-based conservation programs have been implemented around the world since the early 1980s, with significant environmental, social, cultural, and governance outcomes [8]. Environmentally, community-based conservation outcomes include the recovery of wild populations, increased diversity of flora and fauna, an increase in the number of trees planted on private lands, the preservation of critical environments, and the protection of territories comparable in size to traditional protected areas [4,5,6,7,9]. Conservation programs create well-paying jobs for local people and bring a significant amount of capital to communities, allowing them to finance productive projects, provide loans, give scholarships to students, and build infrastructure [4,5,6,7,9]. The cultural benefits of community-based conservation are perceived in people’s attitudes and behaviors towards wildlife which become more reflexive and responsible. For example, the decrease in the removal of carnivores that attack livestock and species that affect crops, in the reduction in domestic use of firewood or the adoption of eco-friendly agricultural practices [4,6,9,10]. In terms of governance, community-based conservation has returned land ownership to local people, strengthened local institutions, improve communication between government agencies and community organizations, and included traditionally marginalized groups such as women in decision-making [8,9].

The community-based conservation programs should considerate several factors that hinder their proper implementation. Songorwa [11] reports that communities are generally not interested in environmental conservation when their interest is temporary and strongly influenced by the promise of economic benefits. In some cases, the economic benefits that communities receive from conservation programs are lower than the benefits they receive from illegal exploitation of natural resources [12,13], or they are not sufficient to cover the costs of damage caused by wildlife, such a crop damage, and livestock depredation [13,14]. Another factor that causes community conservation to fail is the unequal distribution of economic benefits from natural resource use, which occurs when the elites —local or foreign— receive a disproportionate percentage of the profits or when conservation projects employ only a small fraction of community members [5,12].

Community-based conservation begins with an intervention, usually promoted by outsiders (government, NGOs, or academia), that seeks to promote the involvement of local people in the management of natural resources [8]. The intervention must respect the cultural and institutional diversity of the place, integrate traditional and modern knowledge, and encourage reflection on the living conditions of place [7,8,15,16]. In this way, the necessary conditions can be created for community members to take an interest in the conservation project.

The community-based workshops are one of the methods used to implement conservation interventions. These are working sessions in which community members and outsiders share their knowledge and perceptions on a topic of mutual interest through various dynamics [17]. Community-based workshops seek to establish a sincere dialogue between local people and outsiders to create an atmosphere of trust that allows participants to express their opinions, evaluate the knowledge held in the community, exchange knowledge, develop capacity, reach consensus on the interests and needs of the parties involved, and establish the objectives of the conservation program [17,18,19].

Community engagement in biodiversity conservation is of relevance in Mexico, since rural communities hold 32% of the country´s woodlands, forests, and scrublands [20]. Therefore, their involvement in wildlife management is key for the success and continuity of biodiversity conservation initiatives [21,22]. The aim of this research was implementing a workshops program in two rural communities in Northwest Mexico to provide and generate information to propose wildlife species management strategies. The specific objectives of the research were to provide community members with information on the natural history, management, and monitoring of a wildlife species; to document the risk factors that threaten its conservation in the study area; to locate sites in the study area that are important for the target species; and to plan a population monitoring strategy in which all stakeholders would have an opportunity to participate. The research questions posed in this paper are threefold: (1) How should a community workshop be structured to facilitate the exchange of information among stakeholders? (2) What wildlife management tools can be produced in a community workshop? (3) How can formal and traditional knowledge be integrated to develop a wildlife monitoring strategy?

### Case Study: Bighorn Sheep in Two Rural Communities of Baja California, Ejido Cordillera Molina and Ejido Matomí

The study was conducted in ejido Cordillera Molina and ejido Matomí. Ejido Cordillera Molina is an agrarian society constituted by 72 *ejidatarios* (owners of the land), which is located near the border with the United States. The ejido presented a surface area of 141,288.9 ha (https://phina.ran.gob.mx/index.php accessed on 22 March 2022), which is suitable for the bighorn sheep in the Sierra Juárez mountain range [23] (Figure 1). There are fourteen human settlements in the ejido, including ranches dedicated to cattle and equine breeding, ecotourism, mule deer (*Odocoileus hemionus*) hunting, and the exploitation of forest resources (http://www.conabio.gob.mx/informacion/metadata/gis/unikloc10gw.xml?_httpcache=yes&_xsl=/db/metadata/xsl/fgdc_html.xsl&_indent=no accessed on 22 March 2022). The ranches are connected by a network of dirt roads accessed by the Federal Highways 2 and 3 (https://www.inegi.org.mx/app/biblioteca/ficha.html?upc=889463674641 accessed on 22 March 2022; Figure 1).

The ejido Matomí is an agrarian society constituted by 170 *ejidatarios*. It is located on the coast of the Gulf of California south of the port town of San Felipe. The surface area is 224,094 ha (https://phina.ran.gob.mx/index.php accessed on 22 March 2022), which has the environmental conditions for the presence of the bighorn sheep in the Santa Isabel mountain range [24] (Figure 1). This ejido has 30 human settlements, 29 of which are fishing camps mainly dedicated to sport fishing; the remaining is a livestock farm and hunting ranch, in which cattle and horses are raised, and mule deer hunting is practiced (http://www.conabio.gob.mx/informacion/metadata/gis/unikloc10gw.xml?_httpcache=yes&_xsl=/db/metadata/xsl/fgdc_html.xsl&_indent=no accessed on 22 March 2022). The fishing camps are connected by the Federal Highway 5, while the ranch is reached by a dirt road accessed by the same federal road (https://www.inegi.org.mx/app/biblioteca/ficha.html?upc=889463674641 accessed on 22 March 2022; Figure 1).

The reasons why these two sites were chosen to implement the community workshop program are diverse and unique to each of them. The ejido Cordillera Molina is a rural community with which the research team has been collaborating on various conservation projects since 2014. This has created a relationship of trust between the community and the researchers; in addition, the mountain range in which this ejido is located, the Sierra Juarez, is on the international border between Mexico and the United States and is a key point of functional connectivity for bighorn sheep populations [25]. The research team had no previous relationship with the ejido Matomí; however, it was decided to work with this community because the Sierra Santa Isabel, located within the ejido Matomí polygon, is the mountain range with the largest contiguous bighorn sheep habitat in the state of Baja California [24]. 

The bighorn sheep is a resource that has not had a management plan in any of the ejidos for over thirty years, due to the ban that exists in the state of Baja California for extractive exploitation of the species [26]. This ban is in force due to the lack of a management plan that guarantees that extractive exploitation of the species is sustainable in the short, medium, and long terms. This situation has generated serious repercussions for the conservation of the bighorn sheep, as it has caused the abandonment of the species distribution areas by the community that is responsible for protecting them with the consequent proliferation of feral fauna, and lack of control over poaching and illegal extraction of forest resources [27,28].

The greatest challenges to involving these two ejidos, and any other rural community in Baja California, in bighorn sheep management are resentment of the academy and distrust of government institutions. The resentment toward academia is due to the perception in the rural communities of Baja California that academia is the main opponent of bighorn sheep sport hunting. The mistrust of government institutions is due to the fact that in 2010, the Baja California State Government approached the ejidos to develop a project with the promise that the ban on bighorn sheep hunting would be lifted [29], but when the project was completed, the ban continued.

Within this framework, the program of community workshops was developed with the purpose of encouraging the bighorn sheep management in these two ejidos, based on training community members, identifying risk factors for the species, and involving the community in monitoring the species population.

## 2. Materials and Methods

The wildlife program was generated with three community workshops on each ejido from October and December of 2021. In ejido Cordillera Molina, the first two workshops were carried out on the same day (2 October ) and the third one a week after (9 October ). In ejido Matomí, the first two workshops were carried out on the same day (4 December ) and the third one on 11 December. In order to invite the community to participate in the workshops, the research team contacted the ejido presidents and asked that they be allowed to attend a “general assembly of *ejidatarios*,” a meeting held periodically and attended by the majority of the community members. In one of these meetings, the invitation to the workshops was issued, the project’s objectives were explained, the dates and place were agreed upon and those interested in participating were registered. In addition, as recommended by Benchimol et al. [30], it was explained that time spent in workshops would not be paid. However, it was mentioned that organizers would provide meals during each working session.

### 2.1. Setting

The workshops were held at places appropriate for projecting multimedia images. There was a poster placed at the entrance of such sites with the name, the objectives, the topics, and the activities to be conducted, as well as a registration table that served as a sanitary filter to prevent the spread of SARS-CoV-2. The seating arrangement was U-shaped to encourage attendees’ participation, and dialogue between them and with the workshop facilitators [31]. Moreover, participants were encouraged to maintain an open posture (free from crossed arms), because according to Barkai [32], this posture increases the listeners’ attention span. Each participant received a booklet that contained the topics that were going to be addressed during the training workshop and also had space for note-taking.

### 2.2. First Workshop: Training

The first work session, prior to beginning, facilitators were presented, the objective of the workshops was repeated, and facilitators and participants were given the opportunity to set forth the reasons that encouraged them to participate on the project. In this presentation, the facilitators highlighted the ecological, social, and cultural importance of the bighorn sheep in Baja California. They also highlighted the environmental services that the species’ habitat provides to the surrounding towns and cities.

The first workshop was divided into three modules: the first module addressed the natural history of the bighorn sheep; the second module focused on the species management; and the third module discussed the monitoring of the bighorn sheep population. The information flowed from the facilitators to the participants. However, if the facilitators presented data that did not agree with local knowledge, a space would be opened to discuss the issue and clarify the points of disagreement. Similarly, participants were asked to share with the group any additional information they had in addition to that presented by the facilitators. 

The first module informs to participants regarding the most important features of the species natural history, which include its characteristics, distribution, classification, life cycle, behavior, and diet. The information was presented on a slide presentation, which was designed based on the work of Geist [33], Monson and Sumner [34], Rezaei et al. [35], and Valdez and Krausman [36]. At the beginning of this module and with the purpose of capturing the participants’ attention, an image of a bighorn sheep was screened, and the participants were asked to describe the specimen. 

The management module began with an activity in which participants were organized into three work groups, each moderated by a facilitator, and the participants were asked to identify the risk factors and their effect on the species. The work groups chose a representative who presented such information. The risk factors and the effects that were not identified by the work groups, were pointed out and explained by one of the workshop facilitators. The module ended with a learning session that defined the concept of wildlife management, and described the activities normally associated with the management of the bighorn sheep according to Foster et al. [37], Lee [38], Monson and Sumner [34], Smith and Krausman [39], and Valdez and Krausman [36]. The learning session emphasized the benefits of the species and their value as resource if managed well by rural communities.

The last module includes ask to participants concerning about their participation in bighorn sheep monitoring, the purpose of the monitoring, the method used, the materials used, and the results obtained. After, a slide presentation was made to clarify the concept of wildlife monitoring, and to lay out and explain the correct implementation of the methods used to monitor the abundance, structure, and health of the bighorn sheep population; such presentation was prepared based on the work of Burnham [40], Conroy et al. [41], Guerrero et al. [42], and Perry et al. [43].

### 2.3. Second Workshop: Participatory Mapping

The second workshop generated a map of the geospatial information that participants had of the land they share with the bighorn sheep [18,19,44]. Three topics were addressed in this workshop: description of the territory, areas of importance for the bighorn sheep, and risk factors for the species. The description of the territory consisted in naming mountains, canyons, streams, and other structural elements that attendees recognized on the map, and, verifying that all towns, ranches, and vehicular roads were found on the map. The areas of importance for bighorn sheep include located water bodies that used by the species [45] and the areas where sightings of these animals commonly occur were outlined. The risk factors for the species that were pointed out in this workshop were domestic and feral livestock grazing areas [46]; fences, roads, and breaches [47,48]; hiking routes [49]; mines [50]; open-pit dumps [51]; and areas in which poaching frequently occurs [52].

The dynamics of participatory mapping consisted of placing three 35 × 47 inch maps in the center of the room, in which satellite images were shown, as well as the localities (http://www.conabio.gob.mx/informacion/metadata/gis/unikloc10gw.xml?_httpcache=yes&_xsl=/db/metadata/xsl/fgdc_html.xsl&_indent=no accessed on 22 March 2022) and the road network (https://www.inegi.org.mx/app/biblioteca/ficha.html?upc=889463674641 accessed on 22 March 2022) of the northern, central, and southern regions of the ejido at a scale of 1:20,000. Participants placed themselves around the maps, where they pointed out the elements requested by facilitators with permanent markers on an acetate sheet that was placed over the maps.

### 2.4. Third Workshop: Monitoring Strategy 

The third workshop generate a sampling method to carry out community monitoring of the bighorn sheep population. The sampling method was defined by discussion among participants and facilitators, based on the resources available to the community and the characteristics of the land in which the bighorn sheep inhabits, as well the resources as limiting factors to monitor the species. After defining the sampling method, a format was created to register the abundance, structure, health, and distribution of the bighorn sheep populations. The monitoring activities were scheduled, and the responsibilities of the community and facilitators were agreed upon during the implementation of community monitoring.

The field format was tested, where participants were divided into three work groups and each group was provided with a field format and photographs, and videos of bighorn sheep were projected. The teams were asked to register the number of animals that were in the projections, the class, age, gender [33], and body condition [42] of each individual, and the location of the sighting. After this activity, participants shared their opinion regarding the use of the format and indicated the necessary changes to make it easier to use. 

At the end of the workshop, agreements reached in the community workshops were summarized, and a discussion was conducted regarding the future of the species management project and its goals.

### 2.5. Salience Index

A list was created of the participants’ responses to the descriptive characteristics of bighorn sheep. In the list, the responses were separated by participant and the order in which they were expressed was maintained. The salience index [53] of the responses was calculated to determine the importance of these attributes in the rural communities’ perceptions of bighorn sheep.

The salience index of the lists of bighorn sheep risk factors developed in the management module of the first workshop was calculated. Based on the results of this analysis, the risk factors most important to rural community members were determined.

### 2.6. Community Ivolvement

Community involvement in the Bighorn Sheep Management Project was determined by their participation in population monitoring of the species. The use of field formats and the participation of community members in monitoring expeditions were evaluated. The number of sheep recorded in the field formats per participant, the number of sheep observed in each encounter recorded in the field formats, and the number of people who participated in the monitoring expeditions were compared between the two ejidos. We also compared the number of records obtained from direct encounters with the species with those obtained from social media. The nonparametric Mann–Whitney test was used for statistical analysis because the data were not normally distributed.

## 3. Results

The participants that attended the workshops were 37, 21 from ejido Cordillera Molina and 16 from ejido Matomí, who indicated that among the reasons for their participation were the economic interest they hold in the species, as well as the abandonment of the animals, the incidence of poaching on their land, and the fact that this project represents the first approach they had with the academia to work for the conservation of the bighorn sheep and promote the development of their communities.

### 3.1. Community Knowledge 

Attendees at the community workshops describe the bighorn sheep as a wild animal, with large, curled horns, wooly, noble, docile, and territorial, which has many needs, requires a lot of care, and is poorly maintained in the state of Baja California (Table 1). Regarding the distribution, classification, life cycle, behavior, and diet of bighorn sheep, the participants in both ejidos did not provide additional input to the information presented by the facilitators. Participants noted that most of the information presented on these topics was new to them.

Participants in both communities agreed that to preserve the bighorn sheep, it is necessary to perform extractive management, as this would generate the necessary resources to carry out activities to protect the species and its habitat, as well as to provide water and food for animals. In ejido Cordillera Molina, one of the participants expressed it as follows: ‘‘If we had a permit to hunt, we would have money to do what is necessary to take care of the species.” Meanwhile, in ejido Matomí someone explained it in a simpler way: ‘‘If it (the bighorn sheep) gives you an important income, you will take care of it.” However, although participants in the workshops expressed that they are highly interested in the extractive management of the bighorn sheep, they also recognized that ecotourism projects that focus on the sighting of these animals are a good alternative to take economic advantage of the species.

In ejido Matomí, one point of controversy was the impact of burros on bighorn sheep conservation. When the facilitators pointed out that managing bighorn sheep habitat meant eradicating feral burros, one of the participants disagreed. He argued that burros are very important to the ranches in Baja California because they serve as pack animals and as food, so they should not be removed from natural areas.

In ejido Cordillera Molina, participants reported that they monitored bighorn sheep in 2003. They stated that they used binoculars, spotting scopes, and GPS. They also pointed out that in doing this work they learned to look for the animals and classify them by sex and age. When asked what method they used to monitor the population of the species, they said that the method consisted of traveling the mountain range in search of herds and counting the individuals they observed.

### 3.2. Risk Factors for the Bighorn Sheep

The workshops identified 11 factors that threaten the bighorn sheep in the study locations since they cause the reduction in their population and the loss and fragmentation of their habitat. On both ejidos, a total of nine factors had a negative effect, while fires and drug cultivation appear to be relevant only in ejido Cordillera Molina as they were not mentioned in ejido Matomí (Table 2).

Participants reported that fires, drug cultivation, mining, and open-pit dumps, a significant amount of bighorn sheep habitat is lost in their lands. They also indicated that off-road races and ecotourism contribute to this habitat loss, since these activities bring crowds to the mountains, causing contamination and adding new roads and trails to the area.

On the other hand, highways, roads, and fences affect the bighorn sheep’s habitat and make it difficult for the animals to move through the territory. They also pointed out that these structures represent a risk for the bighorn sheep by causing severe injuries, such as being run over or trapped between the barbed wires of the fences.

The biggest threats for the bighorn sheep on their lands, according to attendees, are poaching, livestock, and misinformation from authorities and the society in the state of Baja California regarding the hunting of the species. Poaching, as they said, is an activity that causes death not only of adult male sheep, but also of females and lambs. Regarding this activity, they identified that it is performed on their properties by national hunters, organized crime members, and army personnel. As for livestock farming, participants indicated that it is an activity known to drive the species away from their land due to the fact that livestock compete with sheep for food, water, and space. In addition, it was mentioned that livestock represents a source of diseases for the bighorn sheep. In relation to misinformation in Baja California regarding hunting exploitation of the species, it was considered that this is the reason that prevents them from generating the economic resources required to implement the necessary measures for the conservation of the species in their ejidos. In their view, this problem is caused mostly by the fact that state authorities and the community in Baja California only focus on negative aspects of sport hunting, and do not consider how costly the protection of the vast territory populated by the bighorn sheep is.

### 3.3. Territory Description

The participatory mapping workshop in each ejido named twenty-five structural elements found within the bighorn sheep habitat, which include: mountains, mountain ranges, canyons, and streams (Appendix A: Participatory map of ejido Cordillera Molina; Appendix A: Participatory map of ejido Matomí). The participants of ejido Cordillera Molina reported fifteen natural watering areas and two artificial drinking stations accessible to the species (Appendix A: Participatory map of ejido Cordillera Molina). In ejido Matomí, a total of five watering areas and three oases used by the bighorn sheep were present (Appendix A: Participatory map of ejido Matomí).

In ejido Cordillera Molina, participants noted that sheep flocks move all over the canyon area in Sierra Juarez. Regardless, they also noted that the El Tajo, Guadalupe and Palomar canyons are of great relevance to the species due to their year-round running waters and the fact that the majority of the species’ sightings occur between La Rumorosa highway and Las Palmas military checkpoint, as well as Cañón de los Llanos, because of the high number of travelers in those areas.

In ejido Matomí, participants noted that sheep sightings in their lands occur along the highway; nonetheless, they admit most of these are concentrated in front of the Cinco Islas field and the Miramar and El Huerfanito creeks. Moreover, they mentioned that one of the most important areas for the species in the ejido can be found around the Matomí ranch.

In ejido Cordillera Molina, participants outlined the desert plains near the sierra’s slope as the domestic and feral livestock grazing area. They also indicated that the only road that goes through the bighorn sheep’s habitat is Federal Highway number 2. Moreover, they traced four hiking routes which go through the species habitat, and they located three stone material mines. Finally, they identified an open landfill, as well as two areas where the incidence of poaching is frequent (Appendix A: Participatory map of ejido Cordillera Molina).

In ejido Matomí, they mentioned that the cattle on their land are located near the Matomí ranch, which is the only one dedicated to cattle ranching. Participants also noted that there are four roads that go through the bighorn sheep’s habitat: Federal Highway number five, the dirt roads ranging from San Felipe to the Matomí ranch and from Cataviña to the Santa María Mission, and the Matomí Creek, which is used to move from the Letty field to the Matomí ranch. They also noted that the main hiking route in their territory is the one that goes from the Santa María Mission to the Las Arrastras viewpoint. They marked a mine, from which they said stone materials are extracted, an open landfill and three areas where poaching is frequent (Appendix A: Participatory map of ejido Matomí). 

In addition to the information requested during the participatory mapping workshop, attendees considered it important to indicate the location of water wells in their territory. In ejido Cordillera Molina, they also marked the sites of archaeological importance within their territory, and in ejido Matomí, participants included the polygons of co-ownerships in the ejido.

### 3.4. Monitoring Strategy

In ejido Cordillera Molina it was concluded that the most feasible way to monitor the bighorn sheep population is through trail cameras placed in water catchments, because according to the participants’ words “these are the spots where the bighorn sheep gather”. It was agreed to install cameras on three of the fifteen water catchments of the ejido: El Sapo, La Mora and La Cachucha (Appendix A: Participatory map of ejido Cordillera Molina). It was agreed as well to check cameras every four months. In ejido Matomí it was also agreed that the best option to monitor the species population is through camera trails. They decided to install cameras in the water catchments known as Huerfanito, Miramar and San Luis (Appendix A: Participatory map of ejido Matomí) and check them every four months. The location of the trail cameras was determined based upon accessibility to the places, as well as their distribution along the ejido and the probability of being visited by the species. The timing for checking the trail cameras was suggested by facilitators, based on battery duration and the storage capacity of the memory cards of the devices. 

During this workshop, participants commented to have occasional encounters with bighorn sheep specimens while they carry out their everyday activities in their farms and fishing camps or when they move from one place to another along the ejido. They also indicated that some sightings of the species happen on highways or trails used by people who practice outdoor sports, who usually share photos on social media. In order to make the monitoring more complete, in both localities they agreed to keep record of those occasional sightings.

Participants from both localities committed to purchase the trail cameras, install them, and check them every four months to monitor the population of the species. They also agreed to keep record of all the sightings there might be of bighorn sheep and investigate about the sightings that take place on their lands and get posted on social media. On their behalf, facilitators committed to direct the monitoring during the first year and analyze the results to deliver a report on the state of conservation of the bighorn sheep population in both ejidos.

The field format elaborated to keep record of the community monitoring data, consists of two elements: a twenty-page booklet where they will write the details of every observation (Appendix A: Booklet for keeping record of the sightings of bighorn sheep specimens: (a) cover page; (b) back cover page; (c) back of front page; (d) interior of the back cover page; (e) front page; (f) back of the front page; (g) front of the registration page; (h) back of the registration page) and a leaflet showing the location of the sightings (Appendix A: Leaflet to mark down the location of sightings of bighorn sheep: (a) front; (b) back).

On the booklet, they will write the date and time of the sighting, the number of animals observed, gender, age type, body condition, and the activity of each individual at the moment of the sighting. The registration pages also have a space in which the observer can write any detail that catches his attention during the sighting. Upon request of the attendees, the logo of each ejido was printed on the back of the cover, and graphic elements were added to help mark gender, age type as well as the body condition of the sheep.

On the front page of the leaflet, there is a photographic guide to identify gender, age type and body condition of the sheep. On the back of the front page, there is a map of the ejido on which they will mark with a number the observation, the place of the encounter with the animals. Under the map, there is a space to write down the sighting date, which will help link both parts of the field format.

In the workshops held in both ejidos, it was decided that, in order to have a wider area to register sheep sightings, besides including the ejido’s polygons and the names of the most important mountains, canyons and streams, to also add elements that could only be useful as spatial reference, such as settlements, water catchments, oasis, wells, watering holes, mines, highways, and paths. It was also agreed to add icons easy to identify for the user, such as houses for settlings, the symbol of water for water catchments, palms for oasis, wells for wells, a drop of water for watering holes, and picks for mines. 

### 3.5. Future Perspective

During the final discussion of community workshops in the ejidos, it was agreed that the aim of the project would be to promote responsible use of the bighorn sheep. The elaboration of a management plan of the species based on the results of community monitoring was set as the main goal. The key components of this management plan will be the mitigation of risk factors that threaten the bighorn sheep, the implementation of an ongoing population monitoring program for the species, and the sustainable use of the resource to promote community development. It was also agreed that the communities would fund the development of the management plan and that the facilitators would seek funding to support the process.

### 3.6. Community Involvement

In the study sites, 51 field formats were distributed to record occasional sightings of bighorn sheep, 27 in ejido Cordillera Molina and 24 in ejido Matomí. In ejido Cordillera Molina, five people reported sightings of the species and in ejido Matomí, four people reported sightings of bighorn sheep. In both communities, participants reported encounters with the species that they had personally experienced and learned about through social media (Table 3).

The average number of sheep records generated per participant was 0.19 in ejido Cordillera Molina and 0.63 in ejido Matomí. There was no difference between the number of sheep records generated per participant in the two sites (Mann–Whitney U = 309.5, *p* = 0.68). There was also no difference between the number of sheep observed in each encounter (Mann–Whitney U = 22, *p* = 0.18) in ejido Cordillera Molina ($\bar{X}$ = 2) and ejido Matomí ($\bar{X}$ = 5). Another item in which there was no difference (Mann–Whitney U = 6, *p* = 0.73) was between the number of records generated by direct sightings ($\bar{X}$ = 4) and those obtained through social media ($\bar{X}$ = 1).

In ejido Cordillera Molina, community members provided seven trail cameras for community monitoring of the bighorn sheep population. The cameras were installed in January 2022 and checked three times during the year: in April, August, and December. In ejido Matomí, the community provided three trail cameras for community monitoring of the bighorn sheep population. Eight monitoring expeditions were conducted in this ejido, seven to check the camera traps (February–August) and one to survey the area around the Matomí ranch (November). In both ejidos, the monitoring team consisted of the research group and community members (Table 4); the community members who participated in the monitoring expeditions paid their own expenses. The participation of the community in the monitoring expeditions was higher in ejido Cordillera Molina ($\bar{X}$ = 7) than in ejido Matomí ($\bar{X}$ = 3; Mann–Whitney U = 4, *p* = 0.01).

During the first expedition in ejido Cordillera Molina, there was an encounter with organized crime. Due to this situation, the cameras were installed in different locations than those agreed upon in the workshop. The new places where the cameras were installed were La Virgen, La Mora, and Las Palmitas.

During the first expedition in ejido Matomí, all the water bodies identified in the workshop were visited. Based on the observations made in the field, the research team decided to change the location of the trail cameras to the sites where more evidence of the presence of the species was observed. The sites where the trail cameras were installed were Tinaja del Miramar, Cinco Islas, La O, and Las Palmitas. Trail cameras monitoring was interrupted in August due to the passage of Hurricane Kay, which swept away the cameras that had been installed.

During community monitoring in ejido Matomí, a local resident reported an incident of poaching near Cinco Islas field. As a result, the community called the research team for advice on how to prevent this type of activity on their land. The suggestion was to post “No Hunting” signs. The community purchased 20 signs with their own funds.

### 3.7. Structure of the Community Workshop Program

Based upon the experience in ejidos Cordillera Molina and Matomí, on Table 5 a plan of three community workshop programs was proposed. These workshops could be held during two weekends, to involve rural communities in wildlife management. The time invested in each workshop will mainly depend on the experience of the participants regarding the management of the target species. The topics and dynamics of each workshop can be modified according to the community’s needs in which they are applied and the characteristics of the species of interest.

## 4. Discussion

Wildlife management workshops often attract a large number of people in rural communities because local people see them as an opportunity to share their opinions, concerns and interests regarding the use and conservation of wildlife species, or to acquire new knowledge that will allow them to make better use of the resources they have in their area, or to engage in dialogue with all stakeholders to find a solution to a particular problem [54,55,56]. In the ejidos Cordillera Molina and Matomí, the workshops were well attended: in the case of ejido Cordillera Molina, 29% of the community members participated in the workshops, while in ejido Matomí only 9% of the community members participated, but these were the leaders of the different groups in the ejido, each representing the interests of at least ten people. In both cases, the participants in the workshops were motivated by their interest in using the bighorn sheep as a means for the protection and development of their communities.

The participatory approach of community workshops offers several advantages for wildlife management programs: it builds bridges between stakeholders, improves understanding of resource–user interactions, raises awareness of the benefits of responsible wildlife management, and promotes cooperative and constructive action among participants [54,56]. However, the success of the participatory process depends on the workshops creating the necessary conditions for participants to be able to reconsider their positions based on the points of view of others, so that even if complete consensus is not reached, agreements are reached on the basic points of wildlife management [56,57,58]. In the workshops held in the ejidos Cordillera Molina and Matomí, the participants were able to be open and reflective to the opinions of others by giving everyone the opportunity to state from the beginning the reasons why they decided to participate in this activity, and by showing that although the motivations of each person were different, they all converged on the same point: the conservation of the bighorn sheep.

Participants in the community workshops in the two ejidos acknowledged that their primary motivation for attending the workshops was economic, as they are interested in utilizing bighorn sheep through sport hunting, an interest that Eaton and Martinez [59] noted has existed in rural communities in Baja California since the 1990s. However, despite the great interest for the extractive use of the species they expressed, they did not know key aspects about this activity, such as the specimens’ value and how to commercialize them. Moreover, although the attendees mentioned several investments they would make for the conservation of the sheep to improve their communities with the profits from hunting, none of both ejidos have a business plan that demonstrates the feasibility of such investments, neither an agreement to guarantee that the money from the extractive use of the bighorn sheep would be used on such activities.

In the community workshops, it was clear that knowledge of bighorn sheep natural history, management, and monitoring in the study sites was limited. As evidence, during the natural history module and the session on learning to manage the species, participants expressed that most of the information presented by the facilitators was new to them, and they did not contribute to the topics discussed. This lack of knowledge is common in rural communities, where the information available about wildlife comes from the interactions and relationships that local people have with it [3]. Therefore, local knowledge focuses on the characteristics that are easier to observe and those that are more relevant to the use of wildlife species: uses, diet, distribution, breeding season, population trends, and threats [60,61,62,63].

In ejido Matomí, the impact of burros on bighorn sheep populations was discussed. One of the participants defended the presence of feral burros in bighorn sheep habitat, even though burros have been shown to be a threat to bighorn sheep populations [46] and that feral burro control is an important measure in bighorn sheep management plans [39]. It is common that not all wildlife information from formal knowledge is consistent with that held by rural communities, as local people’s knowledge is influenced by their needs, management practices, and worldviews [3].

In ejido Cordillera Molina, the bighorn sheep population was monitored in 2003. The people in this community indicated that they knew how to use the equipment used to monitor bighorn sheep, and that they knew how to look for specimens, count them, and classify them. However, they did not know that the method they were using to monitor the species was the line transect method, and that they were not implementing it correctly, because they were not collecting data on the distance or angle of sightings, which is essential information for estimating the size of a population according to the description of this method [40]. The limitation of this knowledge in this community is due to a lack of accompaniment by the professional team that directed the monitoring, since they are responsible for training the local people to apply the methods appropriately [21,30].

The community workshops also proved to be a useful tool to build a general outlook regarding the pressure that bighorn sheep populations suffer, since risk factors and impacts that threaten the preservation of the species were identified in the study locations (Table 2). From the issues identified in the workshops, the main causes of the decline of the species’ population were mining, off road races, ecotourism highways, roads, fences, cattle ranching, and poaching [46,47,48,49,52,64,65]. Open-air landfills, planting of narcotic drugs have not been directly related to the decrease in bighorn sheep population; although these two issues are acknowledged as serious threats to wildlife [51,66,67,68]. In terms of society’s negative opinion of sport hunting, this situation has been identified as a threat to community-based conservation projects in Central Asia, where pressure and campaigns to ban trophy hunting are increasing, despite evidence that the resources generated by trophy hunting in that region help prevent habitat conversion and biodiversity loss, and support conservation efforts in protected areas [9]. 

Nevertheless, not all the factors mentioned in the workshops by the participants pose risk for the bighorn sheep, but on the contrary, they could benefit the populations of the species. For example, wildfires were rated in a negative way in the workshops, but it has been demonstrated that they are necessary to restore their habitat and they also open new areas for the species to inhabit [69,70,71]. 

On the other hand, the participative maps elaborated based on the information generated during the second workshop are important tools to manage the bighorn sheep in the study locations, since besides locating the important areas for the species, they show if these areas are impacted by any of the risk factors above-mentioned (Appendix A: Participatory map of ejido Cordillera Molina; Appendix A: Participatory map of ejido Matomí). In ejido Cordillera Molina it can be considered that the information given during the participative mapping workshop is precise because the areas pointed as important for the species match with the ones identified in Sierra Juárez by Ruíz et al. [23] who identified them through a potential distribution model; while in ejido Matomí, participative mapping allowed to identify important areas for the species that had not been identified in other studies held in mountain range Santa Isabel [24]. 

The strategy that will be applied to monitor the bighorn sheep population in the study locations, borrows elements from the programs called *Citizen Science Projects* and from community monitoring [72], to include participation of different sectors of the community and not only of men of working age, who are the only ones who usually get involved in this type of projects [21,30]. Moreover, the proposed strategy allows that the number of members in the monitoring team is not limited by the amount of people who can attend the monitoring expeditions, since every person who wants to participate in the monitoring will be given a field format so they can keep a record of their encounters with specimens of the species or those of which they may hear that happened within the limits of their ejido.

In the two ejidos, very few people used the field formats (Table 3). This is attributed to the fact that the material for reporting occasional encounters with the species was produced in a printed format, which did not motivate community participation. Modern citizen science programs use digital platforms for people to record their observations [72]. These platforms have demonstrated the ability to engage large numbers of people in citizen science projects, largely because they can be used on a variety of devices, project information is updated in real time, and they encourage competition among users [73].

The information collected on the field forms indicates that it is possible to identify areas of importance to the bighorn sheep and to determine the size and composition of its herds from records of occasional encounters with the species. However, in order to make an accurate analysis of these ecological aspects of the species, it is necessary to have a greater number of records than those generated by the field forms [72,74]. Reporting occasional encounters with the species can be a means of generating relevant information about the bighorn sheep population in the study areas if a means of recording sightings is developed that encourages community participation.

Most people who reported bighorn sheep sightings learned about them through social media; however, there was no difference between the number of records obtained through social media and those generated by direct sightings. This is due to the presence of human settlements in the study areas near areas heavily used by bighorn sheep, such as the Cinco Islas field, which results in frequent encounters with the species. These results suggest that social media can be an important tool to complement bighorn sheep monitoring, just as they are for generating information on other wildlife species [75,76].

The study communities actively participated in the trail camera monitoring of the bighorn sheep population: they invested in the purchase of the necessary equipment to conduct the study and participated in the monitoring expeditions with their own resources, i.e., in ejidos Cordillera Molina and Matomí, the community monitoring was carried out without the help of external funding. This is a great achievement of the management project initiated with the workshops, because in general, in community monitoring, the material and expeditions are financed by government agencies or NGOs, and these initiatives are so dependent on this financing that when the economic support ends, the studies stop abruptly, even those that focus on game species [21,30,77]. However, the enthusiastic participation of the communities in the monitoring is also due to the strong economic interest they have in the bighorn sheep, as it is one of the most valuable wildlife species in the world [38] and motivates people to invest in actions that allow them to promote its use.

Ejido Matomí had the lowest participation of community members in the monitoring expeditions because the ejido authorities assigned the person with the most knowledge of the area to accompany the research team on the expeditions to install and check the trail cameras (Table 4). This decision was made to make the monitoring more efficient because the person assigned to accompany the research team, in addition to being the person who knows the area and the animals best, is also the person who lives closest to the monitored area, which also made it easier to make the monthly trips to check the cameras. Similarly, the lower participation of the community in monitoring the trail cameras cannot be attributed to a lack of interest in the management project, since 16 local people participated in the expedition to Rancho Matomí, and when the poaching incident was reported, the ejido invested in the proposed measure to prevent poaching on their land.

In ejido Cordillera Molina, there was greater community participation in the monitoring expeditions (Table 4). In the first expedition, 15 local people joined the work team, but the encounter with the organized crime group on that occasion discouraged people from participating in community monitoring. The presence of organized crime in the study area not only affected community participation and forced the modification of the monitoring planned in the workshops, but it is also a factor that threatens any conservation initiative to be implemented in the area [78,79,80].

The method used to monitor the bighorn sheep population was based on that proposed by Perry et al. [43], which consists of deploy trail cameras in areas frequently used by bighorn sheep and analyzing the results with the Lincon-Petersen method, with the difference that in the ejidos Cordillera Molina and Matomí, the cameras were placed only in natural water catchments used by the sheep, while in the Perry et al. [43] study, the cameras were placed in areas with and without surface water. This decision was made based on the results of the same study by Perry et al. [43], who found that water catchments were the main congregation sites for bighorn sheep and therefore where the greatest number of records of the species could be obtained. In addition, the use of trail cameras, especially in water catchments, to monitor bighorn sheep populations has been used by other authors with good results [42]. It should be noted, however, that since the trail cameras were not distributed throughout the habitat available to the species in the study areas, the population size estimate obtained will only correspond to the region where the camera traps were placed, i.e., in the ejido Cordillera Molina, only the population in the central part of the Sierra Juárez will be estimated, while in the ejido Matomí, the estimate will be for the population inhabiting the coastal region of the Sierra Santa Isabel.

## 5. Conclusions

The program of community workshops proposed in this document allows the inhabitants of rural communities to learn about the natural history and management of wildlife species, to identify elements that threaten the conservation of these organisms, to design a monitoring strategy of these populations that favors the involvement of different sectors of the community, and to motivate people to participate in a wildlife management project. In addition, through this program of community workshops, it is possible to identify the main interests of the community for wildlife species, to map the knowledge that members of the community have about the territory that the species inhabit, thus creating an important tool for wildlife management, and laying the groundwork for the establishment of a long-term management project.

## Figures and Tables

**Figure 1 animals-13-03171-f001:**
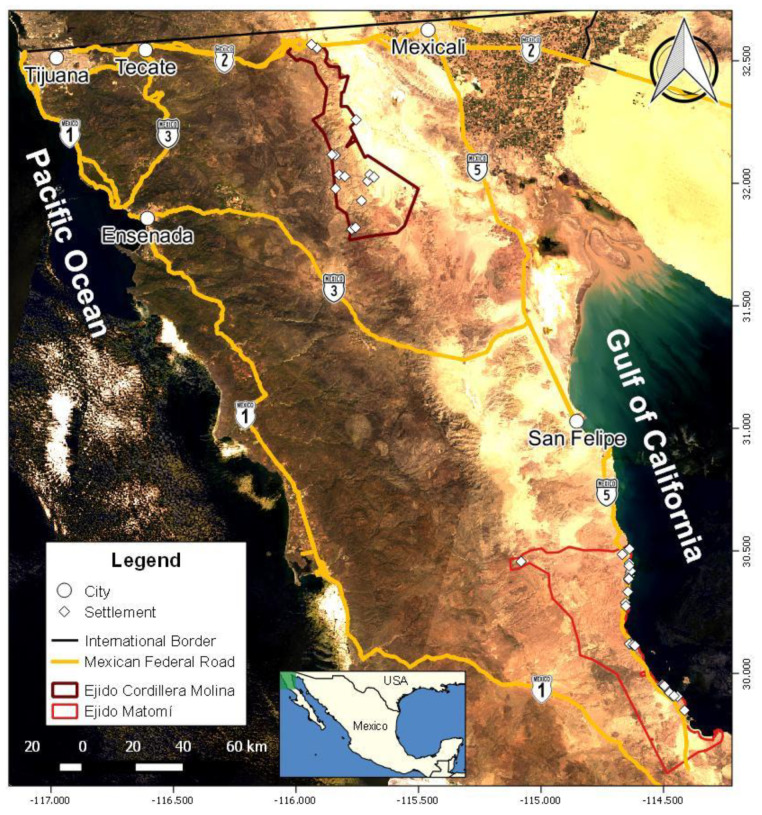
Map of the location of the ejidos Cordillera Molina and Matomí.

**Table 1 animals-13-03171-t001:** Salience index of bighorn sheep attributes identified by workshop participants.

Attribute	Salience Index
Large and curled horns	0.905555556
Wooly	0.851851852
Wild animal	0.781481481
Poorly maintained	0.777777778
Territorial	0.644444444
Which has many needs	0.611111111
Docile	0.444444444
Noble	0.316666667
Requires a lot of care	0.277777778

**Table 2 animals-13-03171-t002:** Bighorn sheep risk factors in research study locations.

Factor	Salience Index	Cordillera Molina	Ejido Matomí
		Impact	
Fires	0.02273	Habitat loss	---
Drug cultivation	0.12175		
Mining	0.14286	Habitat loss	
Open landfills	0.29383		
Off-road Racing	0.22890		
Ecotourism	0.16071		
Fences	0.21104	Habitat fragmentation	
Roads	0.16883		
Poaching	0.34253	Population decline	
Livestock	0.31818		
Misinformation from the authorities and the community	0.22727		

**Table 3 animals-13-03171-t003:** Occasional encounters with bighorn sheep reported by community monitoring participants.

Community	Participant	Recorded Sightings	Date	Place	Sheep Observed	Classification of Sheep Observed			Source
						Ram	Ewe	Lamb	
Cordillera Molina	Alfredo V.	1	20 February 2022	Carretera de La Rumorosa	1	1			Facebook
	Víctor Q.	1	12 March 2022	Cañón de los Llanos	4	4			
	Miguel Q.	1	3 June 2022	Carretera de La Rumorosa	1	1			
	Rogelio R.	1	7 August 2022	Carretera de La Rumorosa	1	1			WhatsApp
	Gerardo V.	1	2 October 2022	Mirador de los Borregos	5	3	1	1	Direct Sighting
Matomí	Agustín R.	11	4 January 2022	Cinco Islas	3	3			Direct Sighting
			20 January 2022		18		18		
			9 February 2022		8	8			
			10 February 2022		7	7			
			23 February 2022		5	5			
			24 February 2022		6	4	2		
			25 February 2022		4		4		
			29 March 2022		1	1			
			13 July 2022		1		1		
			13 January 2022	Miramar	8	8			
			22 January 2022		16	16			
	Daniel P.	1	20 April 2022	Pápa Fernández	2	2			
	José C.	3	1 March 2022	Huerfanito	1	1			Instagram
			9 July 2022	Rancho Grande	2	1	1		WhatsApp
			8 August 2022	Puertecitos	1	1			

**Table 4 animals-13-03171-t004:** Community participation in the monitoring expeditions.

Community	Expedition	Participants
Ejido Cordillera Molina	January	15
	April	3
	August	5
	December	7
Ejido Matomí	February	1
	March	1
	April	1
	May	1
	June	1
	July	1
	August	1
	November	16

**Table 5 animals-13-03171-t005:** Structure of the program of community workshops.

Topic	Objectives	Information	Activity	Time
Introduction	Introduce the technical team that will facilitate the workshops; clarify the objective of the workshops; lay out the reasons that encourage the participation of the technical team and community members.	Situations that motivated the organization of the workshops; ecological, social, and cultural importance of the target species; environmental services that the species’ habitat provides.	Statement of reasons.	~15 min
Training				
Natural History	Communicate the most important aspects of the natural history of the target species.	Characteristics, distribution, types, subspecies, life cycle, behavior, and diet of the species.	Learning session on the natural history of the species.	~40 min
Management	Identify the risk factors that threaten the species; explain what management of the species is about.	Risk factors that threaten the species; definition of the concept of wildlife management; activities comprised in the management of the species.	Work groups to identify the risk factors that threaten the species; learning session regarding the management of the species.	~40 min
Monitoring	Present the methods used to monitor the population of the species.	Monitoring: Definition of the concept; material and methods for monitoring abundance, structure, and health of the species population.	Training session for the correct implementation of the sampling methods of the species population.	~40 min
Territory Description				
Description of the Territory	Building integral knowledge regarding the territory where the species lives.	Structural elements, settlements, and roads that are found in the habitat of the species.	Participatory mapping.	~20 min
Key Areas		Locating bodies of water where the species drink water, as well as its feeding and breeding areas.		~20 min
Risk Factors		Locating elements and activities that fragment, deteriorate, contaminate, or destroy the habitat of the species.		~20 min
Monitoring strategy				
Sampling Method	Establishing the sampling method to be implemented to monitor the population of the species; scheduling of monitoring activities; outlining of responsibilities of the technical team and community members; develop an action plan to address the various contingencies that may arise during monitoring.	Resources and limiting factors to monitor the population of the species in the study area; contingencies that may interrupt monitoring.	Brainstorming.	~40 min
Field Format	Create a field format that can be easily used by community members to monitor the population of the species.	Field formats used for formal monitoring of the species; proposal of field format by the technical team	Training session for the correct filling out of the field formats for monitoring the species and testing the operation of the field format proposed by the technical team.	~40 min
Future Plan	Inform participants about the activities that will come after the community workshops.	Activities that will be carried out after the workshops; goals to be achieved following the workshops.	Informative session.	~40 min

## Data Availability

Data are available on reasonable request to the corresponding author.

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
