# Peer review of "Community-Based Workshops to Involve Rural Communities in Wildlife Management Case Study: Bighorn Sheep in Baja California, Mexico"

_animals, 2023, doi:10.3390/ani13203171_

Round 1
Reviewer 1 Report (Previous Reviewer 1)
The authors provided clarifications to questions and made improvements to the manuscript.
Author Response
Thank you for taking the time to review the manuscript. We appreciate your timely comments.
Reviewer 2 Report (Previous Reviewer 2)
All my comments are addressed hence my recommendation is to accept
Author Response
We appreciate the time you took to review the manuscript. Your thoughtful comments have greatly enriched our work. We are very pleased that you would consider recommending our article for publication.
Reviewer 3 Report (Previous Reviewer 3)
This article reported the process and activities in the development a community-involved conservation and management plan for the bighorn sheep in two Mexican rural communities. Community workshops were used as the main tool to collect local knowledge, encourage local involvement, and establish population monitoring scheme. The article also reported the implementations of bighorn sheep population monitoring activities after the workshops.
Because workshop is a commonly used method worldwide in this kind of occasion, therefore this article requires a discussion on the advantage and disadvantage (such as how to deal with the knowledge gaps on the species between local people and academic facilitators) of the present application, or a discussion on its direct contribution to the conservation of the species in the study area. So far, the only possible contribution produced by these workshops is the bighorn sheep population monitoring project, which including trail camera monitoring and occasional encounter recording. Indeed, a scientific approved and represented monitoring result is the foundation for the sustainable use of natural resources. However, in this study the trail cameras were all installed next the water catchments or water bodies, which covered only one unique habitat type, and thus a significant sampling bias. The validity, as a population trend monitoring method, of the occasional encounter recording is also questionable and require further justification.
Author Response
Dear Reviewer,
We appreciate the time you took to review our manuscript and your comments, which have undoubtedly enriched our work. We would like to respond to your last comments:
- We discussed the advantages and disadvantages of community-based workshops. See lines 528-552.
- Regarding the contribution of the study to the conservation of bighorn sheep in Baja California, we consider that it goes beyond the monitoring project, as the workshops provided relevant information to local people for the management of the bighorn sheep, identified the main risk factors threatening the species in the study communities, produced community maps that are an important tool for the management of the species, and laid the foundation for a long-term bighorn sheep management project.
- The limitations and potential of monitoring the species through occasional sightings is discussed in lines 637-651.
- Water is not a type of habitat; it is a resource, the most valuable one, used by bighorn sheep and distributed throughout the species' habitat. In Baja California, bighorn sheep are found in only one type of habitat: desert mountain ranges.
- We discuss the method we use to monitor the species using trail cameras. See lines 683-698.
This manuscript is a resubmission of an earlier submission. The following is a list of the peer review reports and author responses from that submission.
Round 1
Reviewer 1 Report
The article submitted for review on "Community-based workshops to involve rural communities in wildlife management. Case study: bighorn sheep in Baja California, Mexico," is interesting but needs improvement on some points before possible publication. The idea of joint workshops, uniting communities to cultivate knowledge about the conservation of any breed of animal, much less wildlife, is commendable. My comments-Remove the period at the end of the title. The " Results " section contains figures, the quality and size of which for the viewer is unreadable, this section should be thoroughly improved. The "Discussion" section is too modestly written, also here the authors need to make improvements.
Regards
Minor editing of English language required.
Reviewer 2 Report
What are the specific goals and objectives of the community-based workshops in involving rural communities in wildlife management. How are the rural communities selected and engaged in the workshop process. What are the major challenges faced in involving rural communities in wildlife management, particularly in the case of bighorn sheep in Baja California, Mexico. How do the workshops educate and raise awareness among rural communities about the importance of wildlife conservation and the role of bighorn sheep in the ecosystem. What strategies are employed to ensure active participation and meaningful contributions from the rural communities during the workshops. What are the key components of the wildlife management plan that is developed collaboratively with the rural communities. How are traditional knowledge and practices of the rural communities integrated into the wildlife management plan. What resources, funding, and support are provided to the rural communities to implement the wildlife management strategies discussed in the workshops. How is monitoring and evaluation conducted to assess the effectiveness of the community-based workshops in achieving their objectives. What are the long-term sustainability plans for maintaining community involvement in wildlife management beyond the workshops.
NA
Reviewer 3 Report
This article reported only the process of and activities in the first stage, i.e., communication with the local people, of the development of a community-involved conservation and management plan for the bighorn sheep in two Mexican rural communities. My suggestion is to include some achievements or results to show the significance of the adoption of the plan.